# Novel Planar Strain Sensor Design for Capturing 3-Dimensional Fingertip Forces from Patients Affected by Hand Paralysis

**DOI:** 10.3390/s22197441

**Published:** 2022-09-30

**Authors:** Jacob Carducci, Kevin Olds, John W. Krakauer, Jing Xu, Jeremy D. Brown

**Affiliations:** 1Department of Mechanical Engineering, Johns Hopkins University, Baltimore, MD 21218, USA; 2Department of Neurology, Johns Hopkins Medicine, Baltimore, MD 21287, USA; 3Department of Kinesiology, University of Georgia, Athens, GA 30602, USA

**Keywords:** rehabilitation, hand, finger, stroke, strain, FEA, loading, calibration, plate, 3D, upper-limb

## Abstract

Assessment and therapy for individuals who have hand paresis requires force sensing approaches that can measure a wide range of finger forces in multiple dimensions. Here we present a novel strain-gauge force sensor with 3 degrees of freedom (DOF) designed for use in a hand assessment and rehabilitation device. The sensor features a fiberglass printed circuit board substrate to which eight strain gauges are bonded. All circuity for the sensor is routed directly through the board, which is secured to a larger rehabilitative device via an aluminum frame. After design, the sensing package was characterized for weight, capacity, and resolution requirements. Furthermore, a test sensor was calibrated in a three-axis configuration and validated in the larger spherical workspace to understand how accurate and precise the sensor is, while the sensor has slight shortcomings with validation error, it does satisfy the precision, calibration accuracy, and fine sensing requirements in orthogonal loading, and all structural specifications are met. The sensor is therefore a great candidate for sensing technology in rehabilitation devices that assess dexterity in patients with impaired hand function.

## 1. Introduction

Neurological conditions such as stroke and spinal cord injury can cause significant hand impairment—both in terms of dexterity and strength [1,2,3]. Therefore, there is a need to accurately quantify these two components of hand control.

It has been demonstrated that increasing instructed force in a finger correlates with increasing involuntary force in non-instructed fingers [4]—a phenomenon called enslaving. The ability to suppress enslaving is known as finger individuation [4]. By quantifying the forces exerted by the instructed and non-instructed fingers, an index for assessing finger individuation independently from strength can be calculated. Additionally, strength and precision control of the fingers have been shown to recover via partially separable control systems [4]. This potentially allows for finger dexterity training even in the presence of significant weakness.

Finger individuation is usually assessed in the context of finger flexion [4]. Knowledge of finger individuation in three dimensions is desirable given that dexterity is characterized by flexion/extension/abduction/adduction movements of the finger. Previously developed research devices such as the device by Diedrichsen and colleagues [4] and the HandCARE Device [5] offer force sensing capabilities for the hand, but still cannot capture the rich 3-dimensional workspace of the hand. Devices with multi-DOF sensing such as the Gifu Exoskeleton [6] incorporate more degrees of freedom per finger, but opt to sense finger torque at each joint rather than fingertip force. Thus, fingertip forces can only be estimated, which can result in inaccuracies in kinematic analyses. Additionally, the transmission forces between joints have to be accounted for in a good kinematic model of the hand or else further inaccuracies can develop. Therefore, in order to accurately measure three-dimensional finger individuation, a novel force sensor is needed that is capable of measuring three-dimensional forces at the fingertip regardless of patient strength. To detect very subtle movements in near-plegic hands, this sensor design should also be very sensitive to small forces.

High quality sensing of 3D forces at the fingertip requires fine resolution, low hysteresis, high precision, high linearity, resistance to fatigue effects, low package mass, and low cost. There are plenty of research devices that capture fine forces, but they only detect 1 DOF [7,8,9,10,11,12] and/or do not report detailed sensor performance [13,14,15,16,17,18,19]. There have been several attempts in the academic literature to capture fine 3D force signals from the fingertips with clear performance measures. Artificial fingers have been embedded with small force-sensing packages based on piezo-resistive MEMS technology [20]. Capacitive surfaces have also been attached to the pads of human fingertips to capture pressure and shear contact forces [21]. Additionally, 3D contact forces can be estimated from a photoplethysmograph of blood volume underneath the fingernail [22]. Despite 3-DOF capability and compact sizing, these sensors have limited directional range and resolve down to only hundredths of a Newton. The most suitable sensor used commercially and in academia is the ATI Nano17, which can capture the 3D force at milli-Newton resolution [23]. However, current market price for a single sensor is over $5000, making it fiscally challenging for a five-finger, low-cost home rehabilitation device. Identifying a current gap of implementation in the hand rehabilitation space, we propose a low-cost sensor technology that captures milli-Newton-level forces in three axes (six-directions) with high capacity for strength characterization, and accuracy suitable for integration into rehabilitation therapy devices. Our sensor is based on a strain-gauge approach, given the technology’s prior success in measuring insect and vertebrate biomechanics [24,25] and surgical instrument forces [26]. In the following sections, we describe the design and characterization of this sensor.

## 2. Materials and Methods

### 2.1. Design Requirements

A selective summary of sensors used in upper extremity rehabilitation is detailed in Table 1. In most cases, the sensing technology implemented involved strain gauges or piezoresitive elements.

As described in Table 1, devices in academia and on the market have established sensing capabilities, which we use to inform the specification criteria of our novel sensor. The criteria are as follows:The sensor must fit the fixture geometry indicated in Figure 1. This geometry has a circular area that approximates the cross-sectional area of a fingertip, which is designed to avoid structural collisions with other adjustable finger interfaces (for a multi-finger interface).The sensor, in combination with appropriate amplification and digitization, must sense a resolution of 1 mN or better. For finger force control tasks, oscillatory hand tremor at 10 Hz exhibits typical force power of approximately 0.001 N [29], so this resolution is needed to capture sufficient fine kinematics.The sensor should weigh at most 10 grams so that five would not weigh down their portable chassis in a mutli-finger setup. This is also consistent with the ATI Nano17.The sensor should sense a full-load capacity of 20 N or greater in all spherical directions. Previous literature has determined that fingertip forces can reach up to 60 N for the fingers and up to 110 N for the thumb in dedicated 1D pinch tasks [30], but general 3D finger tasks in a palm-down hand posture report an average finger strength of 20.35 N [4].The maximum accuracy, hysteresis, repeatability, and nonlinearity errors should be at most 5% of the full-scale output. This strikes an appropriate tradeoff from the maximum 3% errors achieved by 1-DOF sensors and is reasonable given that most force sensors for fingertips do not specify their accuracy errors.The sensor should have a maximum displacement of 5 mm at the finger interface, which approximates isometric force sensing.The cost of the sensor from fabrication to assembly should be at most $1000 after economics of scale.

### 2.2. Design Overview

For sensing forces, a planar strain gauge based architecture was opted for due to the technology’s low cost, ease of inspection, and ease of measurement.

FR-4 fiberglass resin was selected as the planar substrate for the sensor board. This substrate choice allows for simplified routing of shorter and smaller circuit traces, as compared to a sheet metal surface, where thicker insulation for wiring is needed. FR-4 also supports power and ground interior layers, making the power layers accessible anywhere on the board, while FR-4 has a much smaller tensile strength and is more expensive than spring metal [31,32], the simplified wiring and resulting smaller assembly volume compared to other implementations outweighs the material’s limitations.

To generate bending loads, the sensor board had to be fixed to mechanical ground. To properly mechanically ground the board, the aluminum fixture, shown in Figure 2, was machined. Holes were drilled in the sensor board to accommodate fastening to the fixture. To transmit the force from a user’s finger to the sensor board, an aluminum frame with a finger-cup interface was secured to the top of the sensor board, while an identical aluminum frame was secured to the bottom of the sensor board (see Figure 3). The finger cup interface consists of a silicone cup attached to a plastic base. A central hole and alignment holes were used to precisely align the frame to the sensor board (using aluminum dowel pins).

### 2.3. Force Sensing Principle

To reduce assembly complexity, all sensing is done on the surfaces of a disk of FR-4 material, which for the purposes of numerical analysis can be modeled as a plate. According to the theory of bending plates, induced bending of a thin beam can be approximated as deforming a 2D plate at the beam’s midplane. The midplane is orthogonal to the plate’s thickness, and the deformation acts parallel to the thickness dimension. The resulting deformation generates strain experienced throughout the plate [33]. Essentially if the strain at select locations on the surface are identified, forces from the fingertip, based on angular vector of deflection, can then be back-derived in 3D space. Strain gauges placed at these unique locations identify the relevant strains by stretching or contracting with the deformation, changing the effective electrical resistance of the gauge. The voltages probed from these resistances can then be mapped to corresponding forces. Since strain gauges work under bending rather than shearing forces, any force component parallel to the plate will not induce strain on the gauges. Therefore, all forces from the fingertip should be applied at a consistent location above the plate via the finger-cup interface. With this design, any parallel force component will cause a bending moment about the center of the sensor, which the strain gauges can detect. To capture three degrees of freedom, at least three strain gauges of unique location and orientation are needed. A fourth gauge was also added to provide information about error by introducing redundancy.

At each planar placement on the disk surfaces, a pair of strain gauges are surface-bonded, one gauge on the top of the sensor board and one gauge underneath, as shown in Figure 4. By connecting the top and bottom strain gauges into a voltage divider circuit as illustrated in Figure 5, the output voltage between the gauges can be determined from a coupled strain-dependent pair of resistance changes. This coupled pair of gauges improves the measurement sensitivity, and increases the resolution of any signal acquired from the sensor. The equation for the voltage divider output is expressed as
(1)$$V_{div}=\frac{R_{G,gnd}+\Delta R_{gnd}}{R_{G,+}+\Delta R_{+}+R_{G,-}+\Delta R_{-}}$$
where $R_{G,+}$, $R_{G,-}$, and $R_{G,gnd}$ are the gauge resistances of the top surface gauge, the bottom gauge, and the gauge closest to electrical ground, respectively; $\Delta R_{+}$, $\Delta R_{-}$, and $\Delta R_{gnd}$ are the resistance changes of the top, bottom, and ground-adjacent gauge, respectively, from applied strain; and $V_{div}$ is the output voltage of the voltage divider.

To prevent voltage comparisons from canceling out during computation of relative *Z*-axis signal later, gauge locations 2 and 4 had voltage dividers with excitation on top and ground on bottom shown in Figure 5, whereas locations 1 and 3 had voltage dividers with excitation on bottom and ground on top in a flipped configuration. As a result, the former configuration has the bottom gauge designated as a ground-adjacent gauge, while the latter has the top gauge as ground-adjacent.

Any change in resistance ΔR caused by an external strain on a single gauge is calculated as
(2)$$\Delta R=GF\times \varepsilon_{S}\times R_{G},$$
where GF is the gauge factor, $\varepsilon_{S}$ is the strain on surface *S*, and $R_{G}$ is the nominal resistance of the gauge package. Therefore, Equation (Equation 1) expands to
(3)$$V_{div}=\frac{R_{G,gnd}+\left(R_{G,gnd}\times GF_{gnd}\times \varepsilon_{gnd}\right)}{R_{G,+}+\left(R_{G,+}\times GF_{+}\times \varepsilon_{+}\right)+R_{G,-}+\left(R_{G,-}\times GF_{-}\times \varepsilon_{-}\right)}$$
where $GF_{+}$, $GF_{-}$, and $GF_{gnd}$ are the gauge factors of the top, bottom, and ground-adjacent gauges; and $\varepsilon_{+}$, $\varepsilon_{-}$, and $\varepsilon_{gnd}$ are the strains applied at the top, bottom, and ground-adjacent gauges, respectively. Any strain gauge on the top surface, $\varepsilon_{+}$, is assumed to experience equal but opposite strain to the bottom surface, $\varepsilon_{-}$, such that $\varepsilon_{+}=-\varepsilon_{-}$. Additionally, all gauges are assumed to have the same gauge factor and nominal resistance. Therefore, Equation (Equation 3) simplifies to
(4)$$V_{div}=\frac{1+GF_{gnd}\times \varepsilon_{gnd}}{2}$$

There are four planar placements where pairs of strain gauges are bonded, specified in Figure 6. Voltage data is interpreted by making comparisons between two different locations. Out of the six unique comparisons, only four are needed to get full voltage information. For example, the comparison signal $\Delta V_{1-2}$ between two given numerical locations 1 and 2, is computed as:(5)$$\begin{array}{c} \Delta V_{1-2}=\left(\frac{V_{exc}\times GF}{2}\Delta \varepsilon_{1-2}\right)\times G+\frac{V_{exc}}{2}, \end{array}$$
where $V_{exc}$ is the excitation voltage, GF is gauge factor, *G* is any gain applied to magnify the difference in divider outputs, and $\Delta \varepsilon_{1-2}=\varepsilon_{1}-\varepsilon_{2}$ is the difference in strain between two locations. The comparison signal voltage should be half the excitation voltage at equal (or zero) strains to minimize saturation of the comparison signal under load. Again, the equations assumes a constant nominal gauge factor, nominal zero-strain resistances across strain gauges, and no thickness change upon deflection.

These comparisons compensate for changes resulting from temperature effects and perform common-mode rejection to eliminate unnecessary direct-current (DC) bias. Therefore, a virtual Wheatstone bridge is created when comparing different placements. The relative voltage signals for each force direction can then be described by the following relationships:(6)$$\begin{array}{c} \begin{array}{c} V_{x}=\Delta V_{4-2}\cos\phi -\Delta V_{3-1}\sin\phi -\frac{V_{exc}}{2}\left(\cos\phi -\sin\phi \right)\\ V_{y}=\Delta V_{4-2}\sin\phi +\Delta V_{3-1}\cos\phi -\frac{V_{exc}}{2}\left(\sin\phi +\cos\phi \right)\\ V_{z}=\Delta V_{3-4}+\Delta V_{1-2}-V_{exc}\\ e=\Delta V_{4-2}-\Delta V_{3-1}+\Delta V_{3-4}-\Delta V_{1-2} \end{array} \end{array}$$
where ϕ is the rotational offset of the quadrilateral pattern of gauges from the -*Y* axis going counter-clockwise on top surface (or clockwise on bottom surface), $V_{exc}$ is the excitation voltage, and *e* is the signed gauge board error. Ideally, gauge board error should always be equal to zero from redundancy, and any non-zero value indicates a board defect. In this application, ϕ is 45°. Therefore, the relative signals in Equation (Equation 6) simplify to
(7)$$\begin{array}{c} \begin{array}{c} V_{x}=\frac{\Delta V_{4-2}-\Delta V_{3-1}}{\sqrt{2}}\\ V_{y}=\frac{\Delta V_{4-2}+\Delta V_{3-1}-V_{exc}}{\sqrt{2}}\\ V_{z}=\Delta V_{3-4}+\Delta V_{1-2}-V_{exc}\\ e=\Delta V_{4-2}-\Delta V_{3-1}+\Delta V_{3-4}-\Delta V_{1-2} \end{array} \end{array}$$

### 2.4. FEA Simulation

To model how strain propagates on the sensor, a static bending finite element analysis (FEA) was conducted using the Simulation package in Solidworks 2020 (Dassault Systèmes; Vélizy-Villacoublay, France). For the bending plate simulation, it is assumed that:The strain gauges and substrate operate within their elastic regions.The gauges are symmetrically and perfectly bonded.The axes of bending deformation align with the central point of the disk.The plate is transversely isotropic, which can later be verified by entries of the material’s stiffness matrix and the calibration matrix. The glass fibers that are cross-woven in FR-4 have dimensions on the same order of magnitude, so the bulk material model is approximately isotropic in any planar direction [34].

To break the simulation problem into simpler sub-problems, a representative sub-unit of the sensor (roughly a fourth of the total disk) was modeled that contained a strain gauge, a pair of fastening holes, and a pair of alignment holes. FEA simulation results are shown in Figure 7. A 2.5 N vertical force (directed down into the plate) was chosen for simulation loading so that this loading scenario could be easily replicated during physical characterization; this force amount is a fourth of 10 N, which is expected to be a typical magnitude of exerted force. The strain field shows compression within the strain gauge area when the vertical force downwards is applied. This compression shortens the gauge length, changing its resistance. Therefore, our FEA modeling is reasonable for strain gauge simulation.

After verifying the sub-unit model, the full disk that approximates four sub-units in a diamond configuration was modeled. Geometry of the disk is shown in Figure 8. Boundary conditions were set by the circumference of the disk as well as the interior holes. Here, 10.0 N was applied down the Z-axis so that each sub-unit would still experience 2.5 N of downwards force. An illustration of reaction forces, applied forces, and strain gauge directions is shown in Figure 9. The resulting Von Mises stress field is shown in Figure 10a, with FEA simulation results of strain summarized in Figure 10b,c for each surface coordinate defined for strain analysis, respectively. Since the analysis force is parallel to the Z-axis, the vertical distance of the finger generating the force from the board does not matter. However, simulating sideways forces from the finger through the interface to the board does need careful consideration of the vertical distance since the interface would be imparting a bending moment on the board. For simulations of X-Y forces that act parallel to the board’s X-Y plane, a moment arm of 10.0 N is applied 15.5 mm from the sensor plane. These simulation results are in Appendix A.

### 2.5. Data Acquisition

Four unique voltage pairs (described in Section 2.3) are each compared and amplified by a corresponding programmable amplifier (Texas Instruments PGA309). For four total comparisons on a single sensor, four amplifiers are needed. The amplifier model has a 100 Hz low-pass filter to remove high frequency noise [35]. The amplifier transforms an input voltage difference into a single output voltage according to the transfer function
(8)$$V_{out}=\left(\left(\Delta V_{in}+V_{c}\right)G_{i}+V_{f}\right)G_{f}\times G_{o}$$
where $V_{out}$ is the amplified output voltage, $\Delta V_{in}$ is the difference between probing voltages of two voltage dividers, $V_{c}$ is the coarse offset voltage, $V_{f}$ is the fine offset voltage, $G_{i}$ is an input gain, $G_{f}$ is a fine gain, and $G_{o}$ is an output gain. The corresponding inverse function is
(9)$$\Delta V_{in}=\frac{\frac{V_{out}}{G_{f}\times G_{o}}-V_{f}}{G_{i}}-V_{c}$$

Table 2 shows the default settings for all parameters. $V_{c}$ is set equal to the zero-load $V_{in}$ via the standard zeroing procedure detailed in Algorithm 1, and $V_{f}$ is set such that a zero-load causes an output voltage of $V_{exc}/2$.
**Algorithm 1:**Procedure for tareing sensor readings via writing amplifier settingsSet tolerances 1,2**for** each finger **do**    **for** each amplifier **do**        Set gains, offsets to default                                                                          ▹ See Table 2        Set counters 1,2 to zero        **while** counter 1 is less than 10 **do**           Obtain 1000 readings           Compute average output reading           Compute input estimate from average output with Equation (Equation 9)           Round input estimate to nearest discrete coarse offset ▹ Refer to PGA309 User’s Manual Table 6–10 [36]           Set coarse offset to rounded estimate           Write coarse offset to amplifier           Compute error between offset and original input estimate           **if** tolerance 1 exceeds error **then**               Break while loop           **else**               Increment counter 1           **end if**        **end while**        **while** counter 2 is less than 10 **do**           Obtain 1000 readings           Compute average output reading           Compute difference between half scale output and average reading           **if** tolerance 2 exceeds difference **then**               Break while loop           **else**               Compute step as negative scaled difference               Compute sum of step and current fine offset               Set fine offset to sum               Write new fine offset to amplifier               Increment counter 2           **end if**        **end while**    **end for****end for**

After amplification and offset, the resulting differential voltages are digitized through analog-digital conversion (ADC) and organized into data packets inside a Teensy 3.5 microcontroller. The microcontroller also conditions the data with a 60 Hz notch filter to remove electromagnetic interference (EMI) caused by electric noise generated from wall power. Upon packet construction, the data with four differential voltages are transmitted to a personal computer (PC), where the data is used to compute relative force signals described in Section 2.3. Afterwards, the relative signals are bandpass filtered between 0.5 Hz and 80 Hz with a 2nd-order Butterworth filter to capture normal human motion as well as stroke-related tremor [29,37]. For summary statistics, mean averaging was utilized.

### 2.6. Sensor Assembly

The sensor assembly is shown in Figure 11 with the finger interface, fixture, and aluminum frames attached to the sensor. (It should be noted for convention’s sake that the upper (top) aluminum frame is connected against the bottom surface of the sensor on the -Z side, whereas the lower (bottom) aluminum frame is fitted against the top surface of the sensor on the +Z side.) The associated free body diagram in Figure 12 demonstrates how applying a fingertip force in the center of the interface generates Z forces and X/Y torques at the sensor surface, which in turn generate bending strain on the sensor surface. Realized assembly photos are displayed in Figure 13. Silicone used for the interface cup was Smooth-On Dragon Skin 10, the plastic cup base was 3D printed from PA 11 nylon, and all structural metal machined was 6061 aluminum alloy. The specific type of FR4 material used is 185HR laminate by Isola Group [31]. The FR4 substrate and copper tracing were provided by Advanced Circuits [38]. The strain sensing packages are S5024 5000 Ω linear gauges provided and installed by Micro-Measurements [39].

### 2.7. Experiments

#### 2.7.1. Characterization

After full assembly, the full-scale output (FSO), sensitivity, and signal-to-noise ratio (SNR) of the sensor were characterized. For this test, a sensor is affixed into a 2-stage assembly of perpendicular rotary tables (see Figure 14 and Figure 15). A collection of calibration weights is incrementally suspended from the sensor until at least one axial voltage reports over 10% non-linearity from a dynamic best fit line. This non-linearity represents breakage of the sensor. The maximum loading weight is the capacity, and the resulting voltage reading is the FSO. The sensitivity of the sensor is then the ratio of voltage to weight (measured using a gram-weight scale) at full capacity. A new sensor is then loaded at 50% capacity and the post-digitization signals are measured for fluctuation. The maximum and minimum levels of the fluctuations are then compared to the FSO to calculate the SNR as follows:(10)$$SNR=10\times log_{10}\left({\left(\frac{FSO}{noise_{high}-noise_{low}}\right)}^{2}\right)$$
where $noise_{high}$ and $noise_{low}$ are the upper and lower voltage limits of the noise fluctuation, respectively.

The resolution of the sensor post-digitization is characterized as
(11)$$res=\frac{V_{FSO}}{V_{exc}}\times \frac{V_{ADC}}{\left(\frac{bits_{ADC}}{bits_{noise,high}-bits_{noise,low}}\right)}$$
where $V_{FSO}$ is the full-scale output voltage of the sensor, $V_{exc}$ is the excitation voltage, $V_{ADC}$ is the operating voltage of the ADC, $bits_{noise,high}$ and $bits_{noise,low}$ are the digital levels of the high and low limits of the noise, respectively, and $bits_{ADC}$ are the digital levels of the ADC.

#### 2.7.2. Calibration

To map the relative axial voltages to real-world force units in all spherical directions, a calibration routine for six orthogonal directions is performed. Orthogonal loading directions were chosen since three bases are needed to fully define a 3D Euclidean workspace. Bidirectional loading was also chosen to validate the sign-flipping symmetry of the workspace. For this test, the same 2-stage assembly of perpendicular rotary tables is utilized (see Figure 15). For each direction, the rotary stages are rotated to their corresponding positions. Due to the specific mounting and markings of the stages, the zero-position of the assembly is actually [270°, 344°], which is the home position for the first and second stages, respectively. To establish orthogonality, each loading direction was a multiple of 90 degrees from the zero-position or “home direction”. The four perpendicular loading directions on the X-Y plane had a 45 degree offset from the canonical X-Y axes to verify that interdependence between the X and Y bases was not present. Furthermore, the two opposite loading directions orthogonal to the X-Y plane were established for Z-axis loading. Table 3 shows the corresponding pairs between loading directions and the angular positions of stage 1 and stage 2. After the sensor is installed in the calibration setup, the sensor is progressively weighed in three cycles of increasing and decreasing loads. A summary of the loading procedure is shown in Table 4.

The x, y, and z components of a weighted loading direction for each trial *i* are calculated as
(12)$$WR_{i}=\left[\begin{array}{c} W_{i}\times \cos\left(\theta_{1,i}-\theta_{1,home}\right)\times \cos\left(\theta_{2,i}-\theta_{2,home}\right)\\ -W_{i}\times \sin\left(\theta_{1,i}-\theta_{1,home}\right)\\ -W_{i}\times \cos\left(\theta_{1,i}-\theta_{1,home}\right)\times \sin\left(\theta_{2,i}-\theta_{2,home}\right) \end{array}\right]$$
where $W_{i}$ is the hanging weight for trial *i*, $\theta_{k,i}$ is the angle of a given stage *k* for trial *i*, and the $\theta_{k,home}$ is [270°, 344°] for stage 1 and stage 2, respectively. After collecting a set of four readings (each from a corresponding amplifier and strain gauge pair) and appending to an array of voltage readings *A* measured during the hanging of weights, the least-squares fit calibration matrix *C* is computed via the Penrose-Moore pseudoinverse
(13)$$C=WR^{T}{\left(A^{+}\right)}^{T}$$
where
(14)$$A^{+}={\left(A^{T}A\right)}^{-1}A^{T}$$
and WR is the vector of x, y, and z loading components.

This calibration matrix assumes that the columns of *A* are linearly independent. Given the large number of readings with measurement noise, it is assumed this is true. However, a rank check on *A* is performed during analysis to verify this assumption.

##### Nonlinearity Error (NL)

Nonlinearity error is the sensor’s inability to output a straight line of readings over the full range of inputs, and is computed as
(15)$$NL=\frac{max|A^{T}-C^{+}WR^{T}|}{FSO}$$
where *A*, *C*, and WR are described in Equation (Equation 13), and FSO is described in Equation (Equation 10).

##### Hysteresis Error (HYS)

Hysteresis error is the sensor’s inability to output the same reading at a given input depending on whether the input falls or rises to that point, and is calculated as:(16)$$HYS=\frac{max|A_{↑,k}\left(W_{i}\right)-A_{↓,k}\left(W_{i}\right)|}{FSO};k=2,3$$
where *i* represents trial index (which also indexes the weight used), *k* represent cycle index, $A_{↑,k}$ represents a reading on the increasing load curve, and $A_{↓,k}$ represents a reading on the decreasing load curve.

##### Repeatability Error (RPT)

Repeatability error is the sensor’s inability to output the same reading upon repeated application of the same input, and is calculated as:(17)$$RPT=\frac{max|A_{↑,k}\left(W_{i}\right)-A_{↑,j}\left(W_{i}\right)|}{FSO};j,k=2,3;j\neq k$$
where *j* represents a cycle index to be compared. *i*, *k*, and $A_{↑,k}$ are described in Equation (Equation 16).

The cycle index *k* indicates the current repetition of the cyclic loading, while the trial index *i* indicates the running count of weights over the whole session. Table 4 details how these indices increment in a loading direction. The reason the first cycle is not considered for error metrics is due to a drifting shift at tare input, known as zero-drift or tare-shift, present from the plasticity of the circuit board.

##### Accuracy (ACC)

Accuracy is the root mean squared of the stated error components, and is calculated as:(18)$$ACC=\sqrt{\frac{NL^{2}+HYS^{2}+RPT^{2}}{3}}$$

Inspecting the relative-force to real-force (3-by-3) calibration matrix reveals the degree of constitutive anisotropy. The more similar the diagonal entries are and the smaller the off-diagonal entries are, the more isotropic the calibration is. However, the more dissimilar the diagonal entries are, the more orthotropic the material is. Furthermore, the larger the off-diagonal entries are, the more anisotropic the sensor is due to strain in one direction generating signals that represent another direction.

#### 2.7.3. Validation

After obtaining an estimation matrix from calibration, it is checked against non-calibration directions to test its ability to predict loads in the spherical workspace from observed voltages. To systematically test the workspace validation, a random set of 30 rotary stage orientations with associated random weights are generated (see example listing of orientations and loads in Table 5). After the sensor is oriented in a given angular configuration, a zero reading is taken followed by the random load reading. The validation procedure then repeats for the next orientation.

The component-wise residuals *R*, which compare the predicted weight components from the actual load components, are given by
(19)$$R=CA^{T}-WR^{T}$$
where WR is the vector of actual load components, *C* is the calibration matrix, and *A* is the matrix of voltage readings from the sensor-ADC system. In terms of percent error, Equation (Equation 19) becomes
(20)$${\left[\%R\right]}_{u}=100\%*\frac{{\left[CA^{T}\right]}_{u}-{\left[WR^{T}\right]}_{u}}{{\left[WR^{T}\right]}_{u}}$$
where ${\left[\cdot \right]}_{u}$ indicates an x, y, or z component of a vector. It should be noted in Equation (Equation 20) that if there is zero anticipated load from regression despite a non-zero actual load, then percent error becomes −100%. If the two load types have inverted signs, then percent error is more negative than −100%.

The vector of system-wise residuals $R_{m}$, which compares overall weight predicted from calibration with actual overall validation weight, are similarly calculated as:(21)$$R_{m}=||CA_{m}^{T}{\left|\right|}_{2}-W_{m}$$
where $A_{m}$ and $W_{m}$ are the voltage readings and applied load at configuration *m*, respectively, and $R_{m}$ is the residual of that configuration *m*. In terms of percent error, Equation (Equation 21) becomes
(22)$$\%R_{m}=100\%*\frac{||CA_{m}^{T}{\left|\right|}_{2}-W_{m}}{W_{m}}$$

Finally, the aggregate root-square of all residuals is computed as:(23)$$R_{RS}=\left|\right|R_{m}{\left|\right|}_{2}$$
where $R_{RS}$ is the 2-norm of the system-wide residuals $R_{m}$.

When plotting the overall residuals and component errors relative to the angular configuration space, surface fits can predict how non-uniform or even non-linear residual errors are depending on the direction of the force vector. Simple inspection of higher-order polynomial coefficients beyond the fitted intercept can reveal predictive non-uniformity or non-linearity, while R-squared coefficients of determination can indicate how well the fits predict the errors.

All fitting curves and surfaces for validation results are generated with the Statistics and Machine Learning Toolbox in MATLAB 2021a (MathWorks; Natick, MA, USA). More generally, all matrix calculations mentioned above are programmatically computed in a MATLAB 2021a script. The script also generates all the characterization, calibration, and validation plotting results below.

## 3. Results

A summary of the sensor’s specifications is shown in Table 6.

### 3.1. Characterization

After fabrication, the sensor unit had a mass of 2.60 g. At breakage, the first test sensor exhibited a capacity of 14.71 N or 1500.0 gf. Effectively, the sensor can read 576.9 times its body weight, a strength-to-weight ratio comparable to a TAS501 S-Type load cell [40]. The sensor also had a maximum full-scale output of 0.289 V. Accordingly, the sensitivity of the device is estimated at a linear 19.64 mV/N or 0.192 mV/gf. The envelope fitting for noise computation is shown in Figure 16. The maximum noise voltage difference of the X, Y, and Z signals was 3.80 mV, which at an ADC reference of 3.3 V and ADC resolution of 16 bits corresponds to 75 digital levels or 6.22 bits. The signal-to-noise ratio was 36.81 dB. The maximum resolution of the sensor-ADC system is 2.564 mN or 0.262 gf.

### 3.2. Calibration

A plot of the calibration voltage response over the loading routine in the +X, −Y loading direction of the test sensor is shown in Figure 17. (Voltage responses for the other five loading directions are shown in Appendix B.) The full voltage response indicates strong repeatability between cycles and strong linearity overall. A selection of three cycles from the first configuration is shown in Figure 18, which identify mild hysteresis in the first cycle and the lack of hysteresis in the second and third cycles. The associated sensitivity matrix for mapping loads to voltages is depicted in Table 7. For this test sensor, the hysteresis distance between loading curves on the same cycle is 2.29% of the full-scale voltage. The maximum deviation from the best fit curve is 2.94% FSO, and the maximum variation of repeated readings is 0.61% FSO. Aggregating these metrics using a root-mean-sum-square formula yields an average accuracy of 2.18% FSO.

### 3.3. Validation

Figure 19 details the loading validation by comparing the predicted axial-component of loading (from the calibration matrix) with the actual axial-components of loading, which are elements of the real number set within plus or minus capacity. Overall composite residuals as a function of overall loading are depicted in Figure 20.

Component-wise, the coefficients of determination from fitting an ideal 1:1 weight prediction curve to the X, Y, and Z components are 88.46%, 96.05%, and 99.12% respectively. System-wise, the coefficient of determination from fitting an ideal weight prediction curve is 96.98%, whereas the coefficient of determination from a best-fit curve is 97.06%. The aggregate root-square of all residuals for the tested sensor was 6.21%. The maximum residual error during validation was 211.4 g from the predicted value of 930.0 g.

Component and overall residual errors as a percentage of applied load are shown within an angular configuration space (or C-space) in Figure 21. The coefficients of the mixed-quadratic polynomial surface fits for each residual error type, as well as associated coefficients of determination, are shown in Table 8. Summary statistics of error by load and a histogram of residual error are provided in Appendix C.

## 4. Discussion

The sensor was successfully manufactured to fit within the fixture geometry shown in Figure 1. The sensor’s mass of 2.6 g after manufacture exceeds the design goal of 10 g, which helps any larger system the sensor would be integrated into stay lightweight and portable. While the final resolution of 2.564 mN does not meet the design resolution of 1 mN, the sensor system is still capable of capturing fine hand tremor signals given that certain tasks only need 20 mN or less of precision [29]. The design result and goal are in the same order of magnitude, and the resolution can be improved with better filtering and higher-bit ADC units. Additionally, while the sensor’s capacity of 14.71 N may have not met the design goal for maximum voluntary contraction measurements, it can capture forces of moderate exertion without any issue. Furthermore, the repeatability error of 0.61% FSO, the hysteresis error of 2.29% FSO, and the nonlinearity error of 2.94% FSO, all do not exceed the design goal of below 5%. This demonstrates that the sensor transduces loads consistently and linearly within and between applications. Combining these errors into an accuracy metric of 2.18% meets the design goal and demonstrates good general sensor accuracy. Due in part to the rigidity of the metal components used to assemble the system, the sensor is able to collect signals without flexing more than 5 mm from neutral state. This was verified with calipers as the sensor was being loaded. Finally, the sensor cost estimate of $350 is well below the stated goal of $1000, making this sensor solution an ideal alternative compared to other systems in the market. Comparisons between the design sensor and current sensors used in finger rehabilitation are given in Table 9. The design sensor is not intended to completely supersede all metrics of any of these sensors and to replace them in the field, but highlight that the proposed sensor design addresses a missing implementation gap in 3D finger sensing. Overall, the sensor’s ability to record 3D forces of small to relatively large magnitude in a light package highlights a great design success.

Despite the general success of characterization and calibration, there are some caveats that warrant discussion. Higher fluctuations in noise and repeatability will affect the system’s ability to distinguish finer motion signals. Stated accuracy metrics from calibration were achieved only after voltage readings from the first cycle of loading were excluded from analysis. Plastic deformation of the tare reading when a configuration is first loaded could explain why the loading curves’ zeroes shift to another configuration. Although hysteresis error within a configuration can meet design requirements, that does not imply the zero-load output will not shift significantly between configurations. The plastic zero-shifting could be resolved by improving the bonding technique and the substrate material, although verification is yet to be done. Additionally, validation of the sensor in the entire 3D workspace of possible applied forces is a more significant challenge. The root-mean-sum-square aggregate of the overall residuals does not meet the target of 5% FSO in the specification criteria. Unusually high residuals may partially be due to the aforementioned plastic tare shifting. Since each of the 30 random readings is its own configuration, a single cycle of loading may be insufficient to fix nonlinearity issues. Another plausible reason is that relative errors from validation loading over the entire C-space is not as uniform as previously understood. This reason is supported by the presence of high error outliers shown in Figure 21. For example, there is a cluster of −100% relative X errors from validating around angular configuration [270°, 254°]. Furthermore, applying surface polynomial fits to the C-space residuals do not exhibit a uniform “horizontal” plane. Instead of just having zero-order affine coefficients, all 2nd-order surface fits in Table 8 exhibit non-zero linear, mixed-interaction, and squared terms. The associated R-squared metrics imply correlations that are weak (component-wise) to moderate (overall). Generating higher-order surface fits that predict residual errors from angular position better would further demonstrate that the C-space is not just non-uniform in error, but also incredibly non-linear. Despite low nonlinearity error being demonstrated across levels of calibration weights, it is still possible for nonlinear residuals to appear across non-calibration loading angles during validation. The introduction of holes in the substrate or the transversely isotropic structure may mechanically explain these non-uniform and non-linear predicted load error trends across the C-space. Furthermore, cross-sensitivity can exist on this sensor due to the lack of mechanical isolation between the strain gauges on the sensor substrate, which may explain the non-zero values in non-diagonal entries of the calibration matrix.

In the future, the dynamics of the sensor should be explored by collecting metrics such as rise/fall time after a step input, vibrational parameters, and fatigue cycles before breakage. Given that human subjects will operate any peripheral device with the sensor in a constantly dynamic manner, understanding mechanics beyond steady-state conditions is important. Furthermore, accuracy errors and residual errors could be improved by either switching the substrate to a very elastic element (sheet metal) with good insulation for signal routing, or by switching to a more robust layout of measuring bending deflection (e.g., capacitive sensing with a stiff dielectric as demonstrated by Jeong et al. [41]). By moving away from a brittle material like fiberglass resin, the sensor should exhibit less variation in plastic tare-shift between configurations and in maximum input capacity. Additionally, we can investigate the effects of cross-sensitivity between gauges with FEA modeling with forces in all axes, and with a semi-supervised method of regressing the calibration matrix rather than a simple linear regression.

Despite these limitations, the presented sensor offers an inexpensive and relatively accurate sensing package for capturing the fine forces generated at the fingertips in all directions. By implementing the sensors in a larger rehabilitative device, dexterous finger control, such as finger individuation and object manipulation, can be assessed and potentially trained independent of hand weakness [4,42]. Precision control can be tracked and improved without needing a separate hand model to estimate task-space forces from joint torques. The possibility of training hand dexterity despite marked weakness increases the chances for patients to regain control needed for activities of daily living (e.g., manipulating and tapping an egg, single-tapping a touchscreen, twisting a dial on a car console, etc.).

## Figures and Tables

**Figure 1 sensors-22-07441-f001:**
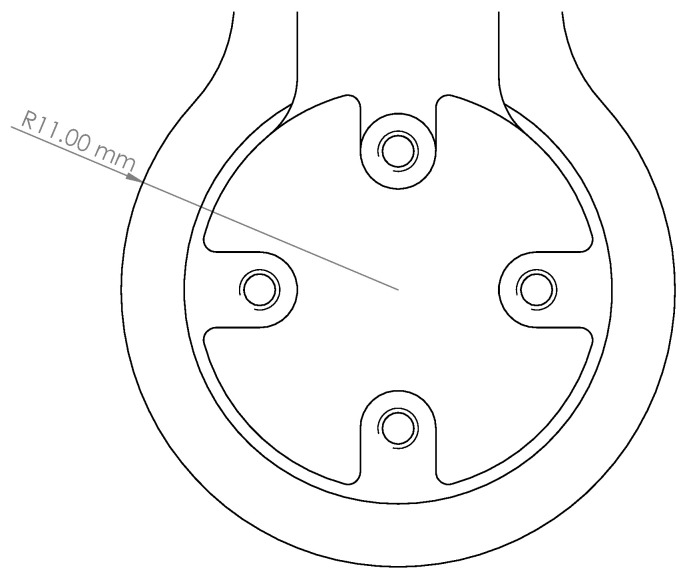
Bounding geometry of the sensor fixture used to fit the sensor to the hand rehabilitation device.

**Figure 2 sensors-22-07441-f002:**
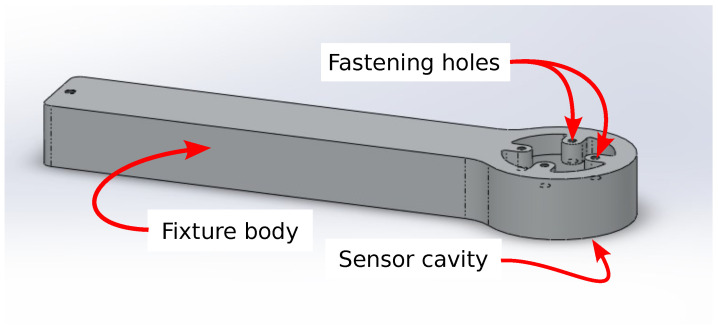
The aluminum fixture that fits and grounds the 3-DOF sensor. The sensor is secured to the fastening holes from within the sensor cavity. The long fixture body allows large-scale adjustment of the sensor to accommodate the user’s unique neutral hand posture.

**Figure 3 sensors-22-07441-f003:**
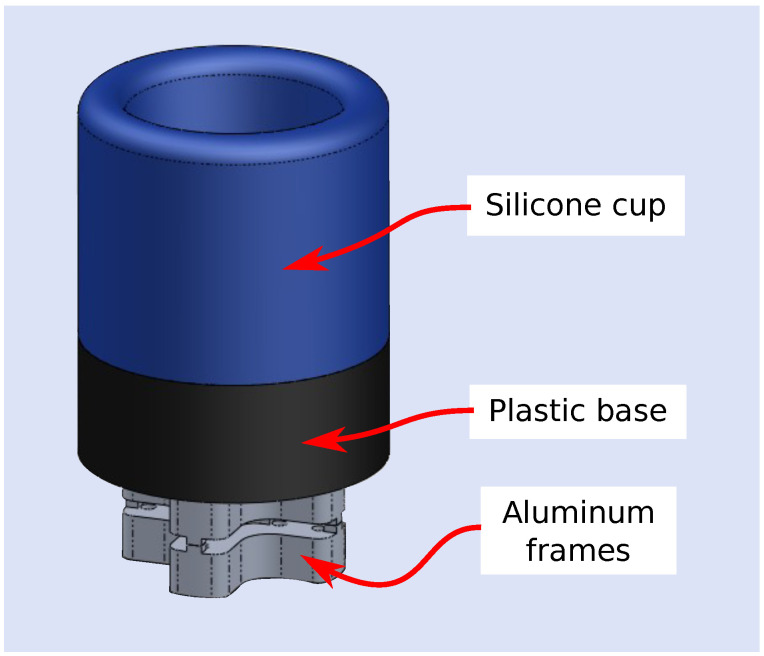
The finger-cup interface attached to a two-part aluminum frame assembly. The 3-DOF sensor fits between the two halves of the frame assembly, which secure together to create a rigid structure on the sensor. A plastic base attaches to the top of this structure to provide a rigid base for a silicone cup. Once secure (using adhesive), the silicone cup provides a comfortable grip onto a user’s finger. The finger forces (and torques) can then be transmitted in total to the body of the sensor.

**Figure 4 sensors-22-07441-f004:**
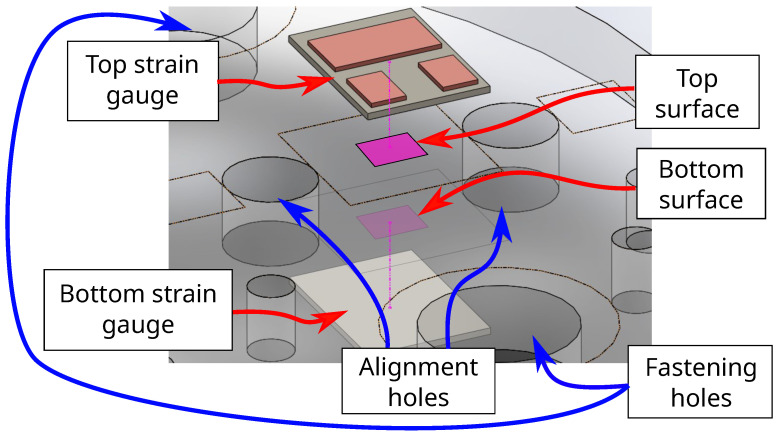
Exploded view illustration of strain gauge units bonding to both sides of sensor board. Stain gauges are placed from above and underneath the sensor body to a location on the surface indicated by the arrows. Holes, for purposes of fastening to the aluminum fixture and for fitting alignment pins between the frame assembly, are rendered in the sensor body for reference.

**Figure 5 sensors-22-07441-f005:**
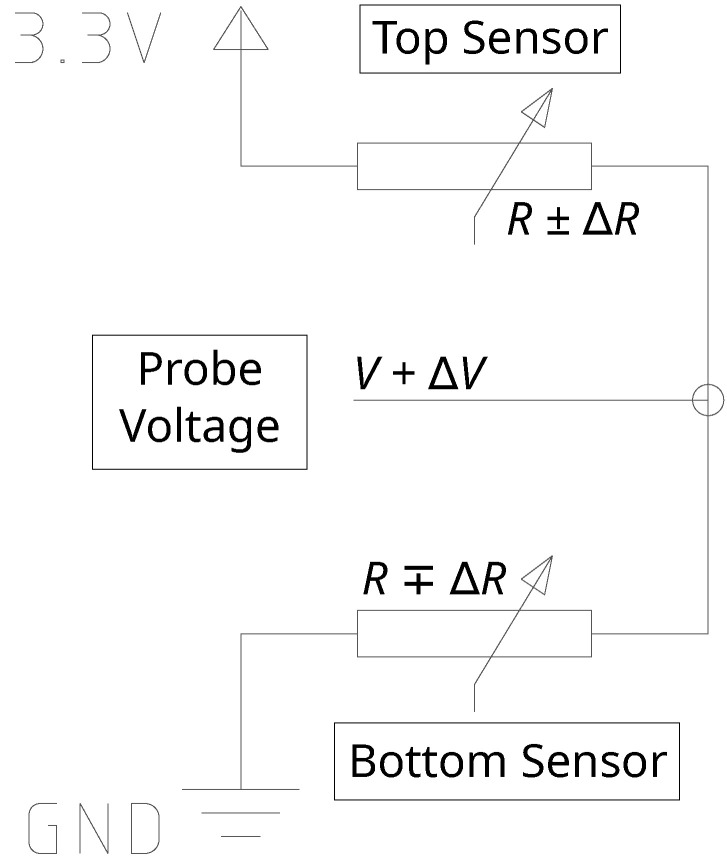
Schematic drawing representing a pair of strain gauges connected in a voltage divider circuit on the 3-DOF sensor. Ideally, the two gauges experience equal and opposite resistance (as shown) resulting in the voltage reading at the probe location having a “doubling” effect compared to only a single-gauge configuration.

**Figure 6 sensors-22-07441-f006:**
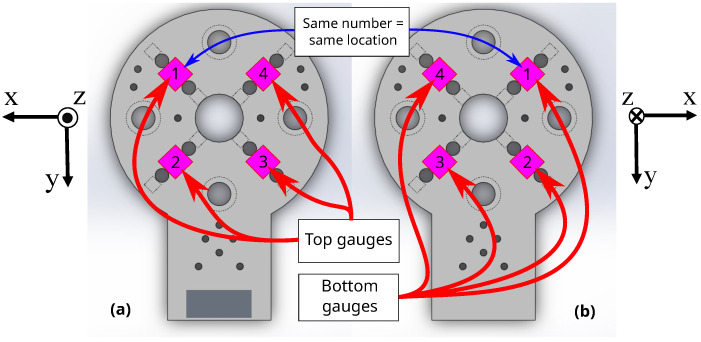
The numerical references of gauge locations 1 to 4 on the (**a**) top surface and (**b**) bottom surface of the sensor. A pair of gauges on top and bottom will share a reference number, but will appear at “mirrored” locations after flipping the body of the sensor. These reference numbers are useful in describing the relationship between Euclidean-space force components and changes in probe voltages between gauge pairs in the voltage divider configurations.

**Figure 7 sensors-22-07441-f007:**
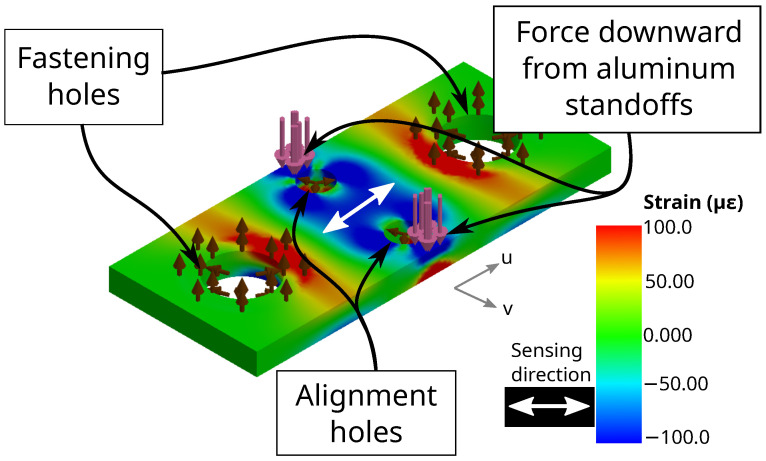
Sub-unit of the 3-DOF sensor with one strain gauge (indicated by the white arrow), two alignment pins, and two fastening holes. A 2.5 N downwards force was simulated, represented by pink arrows at the edge of the sub-unit. Fixed surfaces on the holes are represented by brown arrows. Resulting strain of the sub-unit surface, with respect to the white axis in the u direction, can be negative and compressive (which is colored more blue) or can be positive and tensile (which is colored more red).

**Figure 8 sensors-22-07441-f008:**
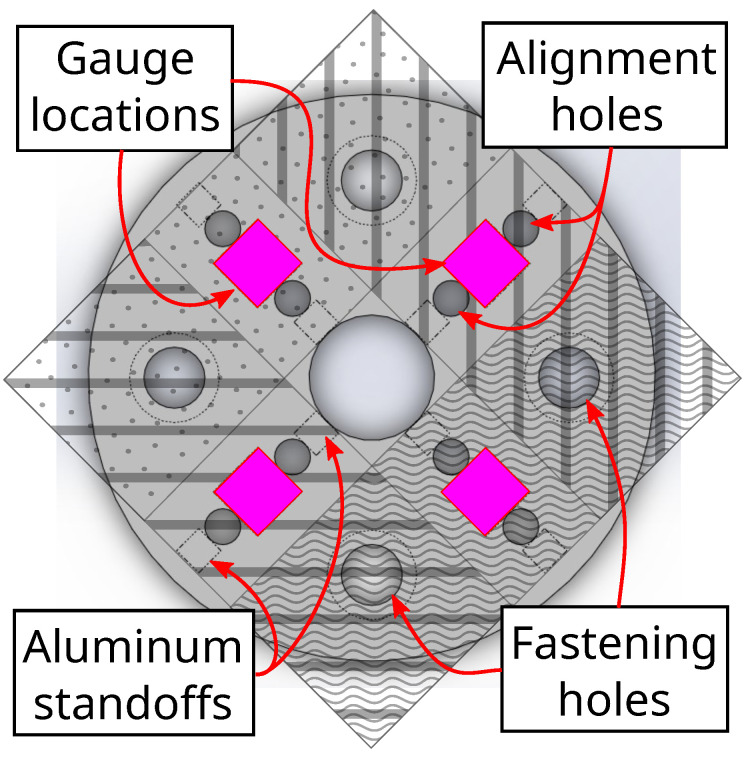
Geometry of disk approximated from four strain sub-units arranged in diamond-shaped configuration. Each sub-unit is distinguished by a diagonal striping pattern; subunit areas are represented by unique rectangular patterns. Pink diamonds indicate strain gauge placements. Alignment and fastening holes are rendered for reference.

**Figure 9 sensors-22-07441-f009:**
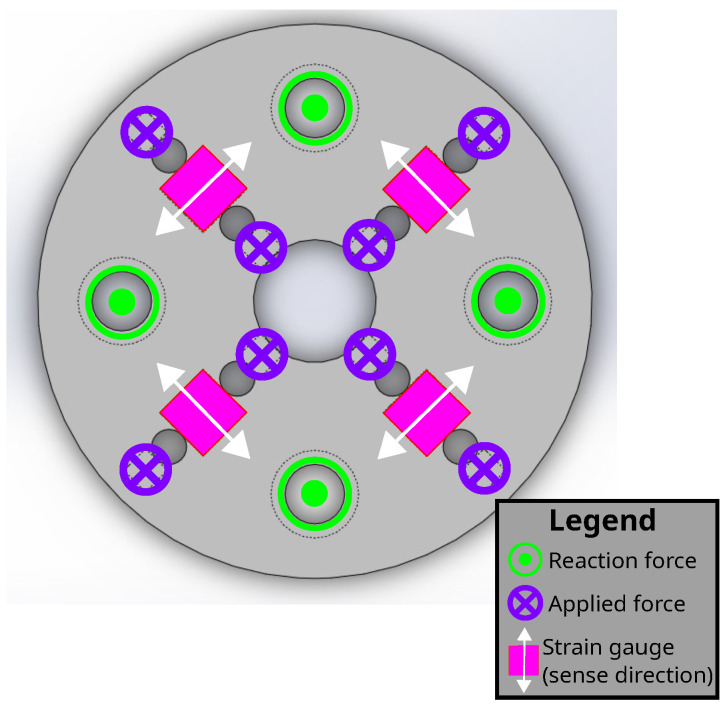
Illustration of applied forces and reaction forces from fixed supports to sensor board, as well as strain-sensing directions. Applied forces in this scenario are all into the page, designated by purple crossed circles. Fixed supports are all out of the page, designated by lime-green dotted circles.

**Figure 10 sensors-22-07441-f010:**
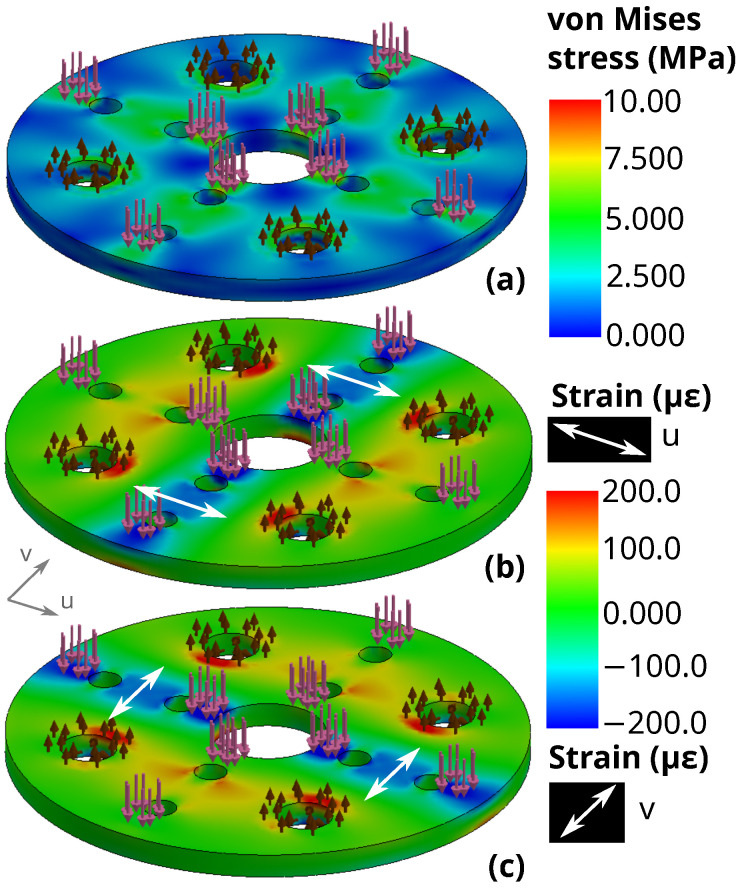
FEA simulation of a vertical 10.0 N downwards force distributed around the strain gauges on the 3-DOF sensor. Applied forces are represented by pink arrows pointed down, while support fixtures are represented by brown arrows. Surface coordinates u and vs. are represented by a grey frame. All results are viewed from top surface. Results include (**a**) von Mises stress field, (**b**) u-direction strain, and (**c**) v-direction strain fields. The axis being referenced for strain measurement is indicated by a white bidirectional arrow on the surface of the sensor at the gauge locations. The two strain axes are perpendicular, but are offset from established X-Y axes by 45 degrees.

**Figure 11 sensors-22-07441-f011:**
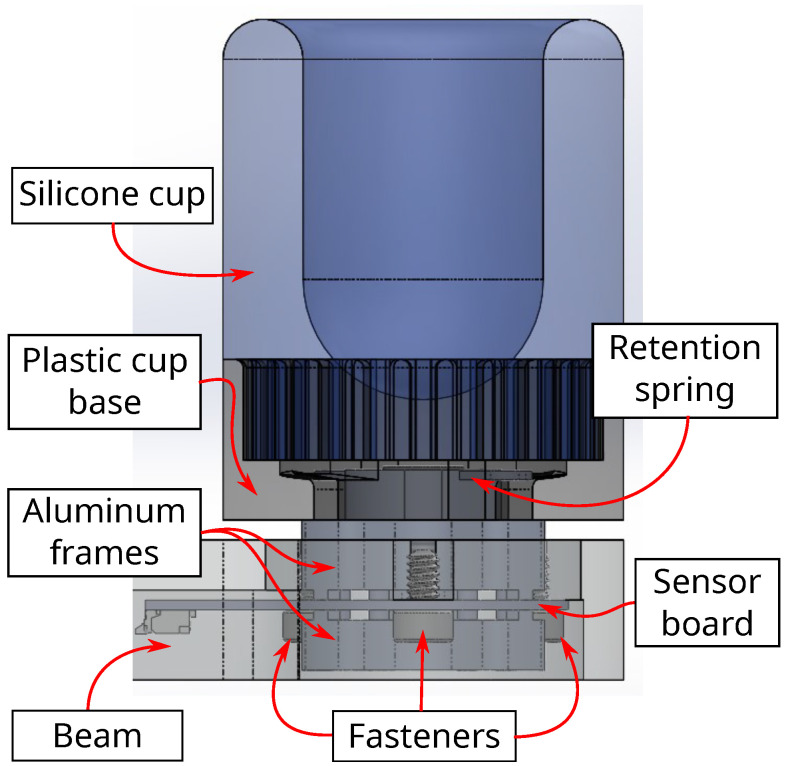
Side view of final assembled sensor-fixture-interface system. The 3-DOF sensor fits and aligns between the aluminum frames, and is fastened inside the beam structure. The plastic base holds the silicone cup for human interfacing, and is held against the top aluminum frame by a retaining spring.

**Figure 12 sensors-22-07441-f012:**
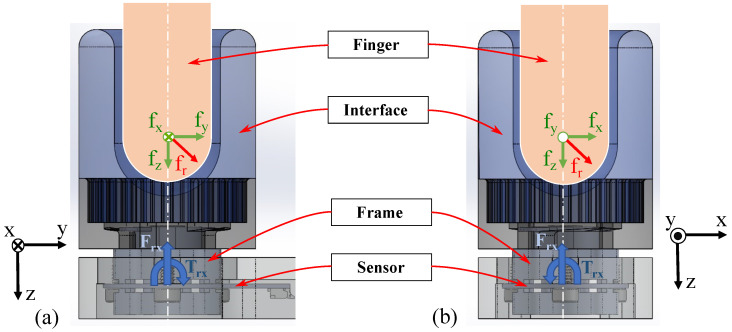
Free-body diagram of the human finger and the 3-DOF sensor within the overall assembly, viewed from the side (**a**) and front (**b**). Total force exerted by the finger onto the assembly is broken down into axial components, designated by green arrows. The resultant finger force is represented by a red arrow. The axial directions are indicated by black coordinate frames to the side. Reaction force and torque are also indicated at the sensor by blue arrows. The reaction components contribute to bending of the sensor, which are sensed by the gauges.

**Figure 13 sensors-22-07441-f013:**
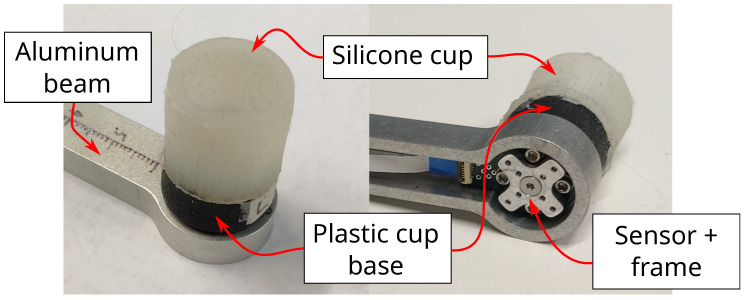
Realized implementation of overall finger assembly viewed from top and underneath.

**Figure 14 sensors-22-07441-f014:**
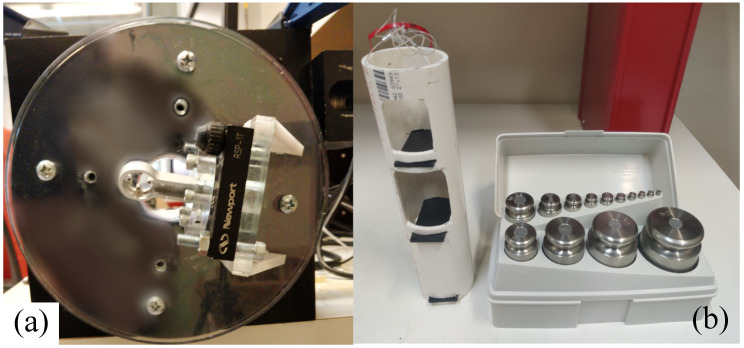
The loading materials needed to characterize, calibrate, and validate a 3-DOF sensor: (**a**) 2-stage rotary table setup for configuring 3D angular orientation of sensor; (**b**) set of loading weights (ranging from 1 g to 2 kg) and loading tube for holding weights and providing line of tension from the sensor.

**Figure 15 sensors-22-07441-f015:**
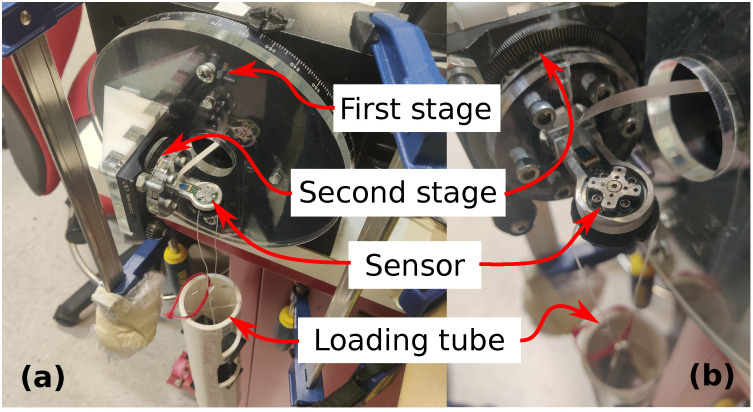
Assembled two-stage loading setup with (**a**) overall view and (**b**) zoomed view of sensor installed in second stage. The second stage is mounted perpendicular to the first stage. The finger assembly is mounted to the second stage such that the sensor’s Z-axis is pointed perpendicular to the normal axis of the second stage. The tension line of the loading tube is secured to the top of the sensor using a custom plastic fastener.

**Figure 16 sensors-22-07441-f016:**
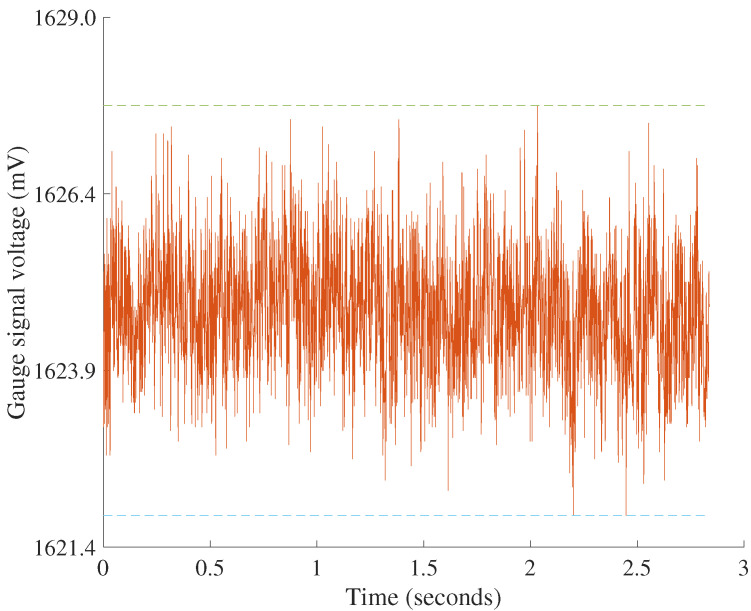
Voltage response of one voltage divider unit during constant 50% loading capacity. A fitted envelope has a green upper bound at the highest noise level and a blue lower bound at the lowest noise level. The height of the envelope represents the magnitude of noise in the signal, which here is 5.79 mV. Note that the maximum noise of 3.8 mV when calculated from X, Y, and Z signals is lower than noise calculated from raw divider signals due to how X, Y, and Z voltages are computed from raw voltages.

**Figure 17 sensors-22-07441-f017:**
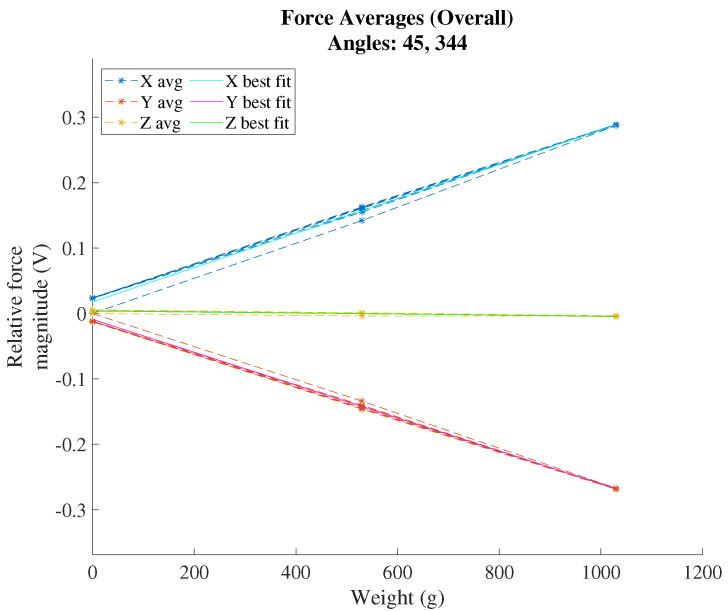
Voltage response to 3D calibration loading in the +X −Y direction. All three forward-backward cycles are plotted in dashed lines, and corresponding lines of best fit are solid.

**Figure 18 sensors-22-07441-f018:**
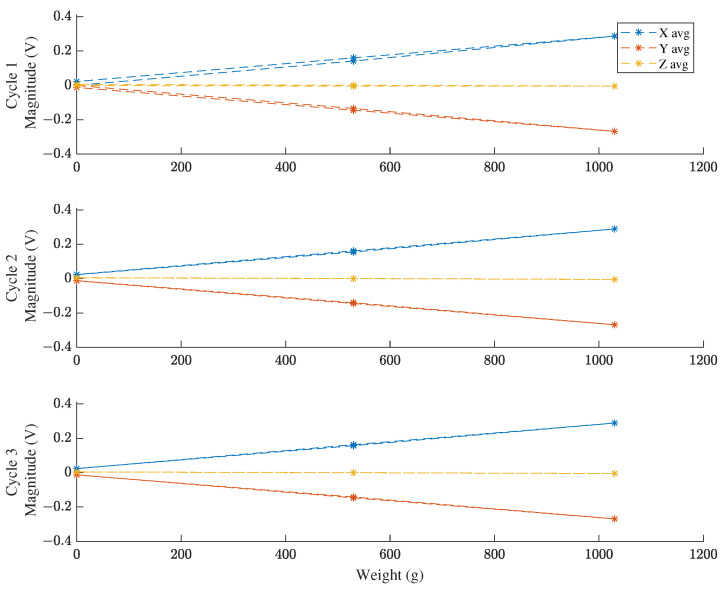
Voltage response to 3D calibration loading in 3 separate cycles in the +X −Y loading direction.

**Figure 19 sensors-22-07441-f019:**
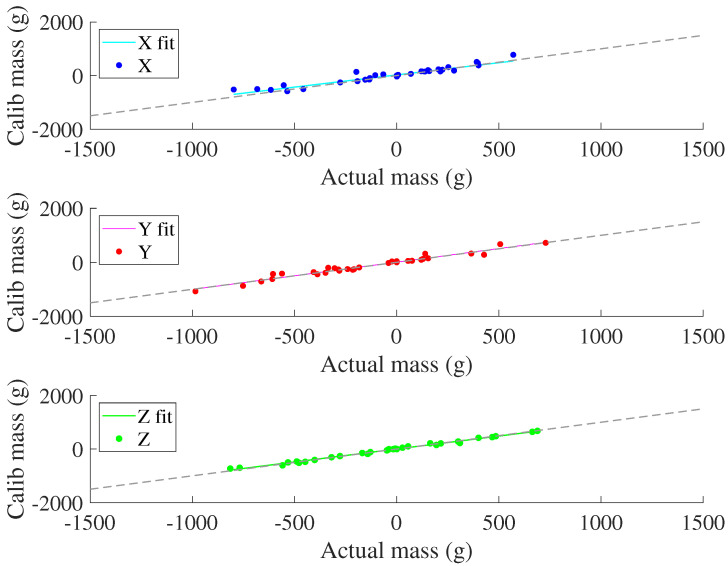
Axial components of actual loading compared with loads predicted by the calibration matrix. The black line indicates ideal estimation matching with actual load.

**Figure 20 sensors-22-07441-f020:**
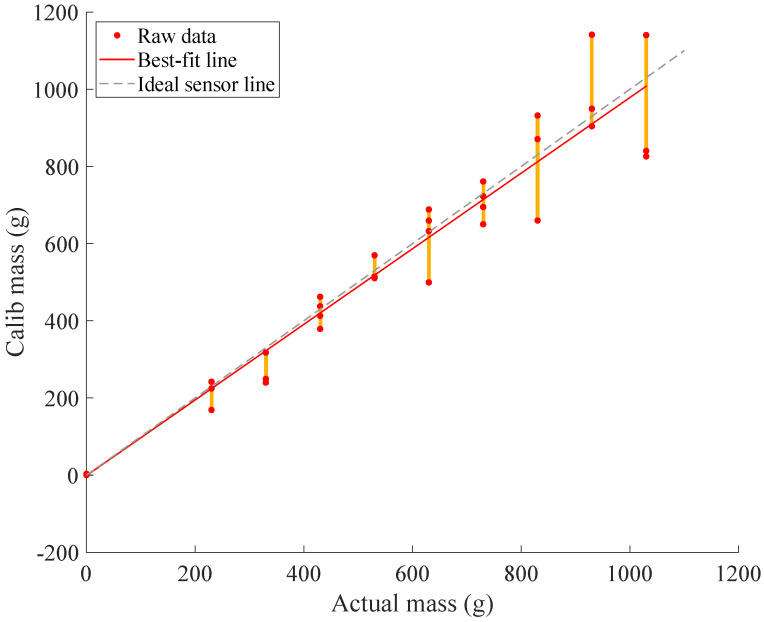
Overall loading residuals as a function of the overall loads applied. The discrete levels of loading are highlighted as orange-yellow vertical lines. The dashed grey line indicates zero error and that actual overall mass matches what is predicted by calibration. Meanwhile, the red diagonal line indicates the line of best fit.

**Figure 21 sensors-22-07441-f021:**
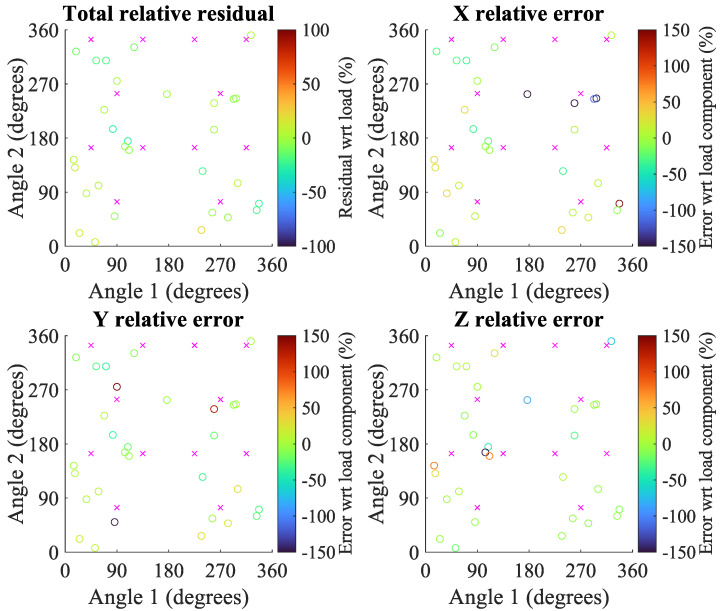
Relative percents of overall loading residuals and axial-component errors as a function of C-space angular orientation, normalized to either the total applied load or respective applied load component (x, y, or z). Magenta crosses indicate the positions at which the sensor was calibrated (with mirrored counterparts), while the circles indicates at what angular positions the errors were sampled and how large they were (indicated by color) for the component or system. Relative component error scaling is limited to 150% in magnitude for easier interpretation of color differences at smaller errors. Component errors more negative than −100% indicate actual and anticipated loads used to calculate error have inverted signs.

**Table 1 sensors-22-07441-t001:** Tabular summary of a selection of sensors used in upper-limb rehabilitation.

Metric	Unit	Diedrichsen [4]	HANDCare [5]	FFPO [27]	MESA+ [28]
Sensor		Honeywell FSG15N1A	Angst-Pfister MilliNewton	ATI Nano17	Wafer ring
DOF		1	1	6	5
Mass	g	3	N/A	9.07	5.93
Span	N	15	0.4 to 2	12	60
Accuracy	% FSO *	2.25	1	N/A	N/A
Hysteresis	% FSO *	3	N/A	N/A	N/A
Nonlinearity	% FSO *	1.5	N/A	N/A	1
Repeatability	% FSO *	2	1	N/A	N/A
Cost	$	118	N/A	5000	N/A
Planar dim	mm	12.7 × 18.2	25.4	17	9
Depth	mm	9	12.7	14.5	1

N/A: Specifications are not publicly known. * FSO: Full scale output.

**Table 2 sensors-22-07441-t002:** Default parameter settings for the PGA309 amplifier transfer function (see Equation (Equation 8)) converting a pair of probing voltages from a gauge bridge into a larger output voltage.

Setting	Value
$G_{i,0}$	32
$G_{f,0}$	1
$G_{o,0}$	9
$V_{f,0}$	3.33/18
$V_{c,0}$	0

**Table 3 sensors-22-07441-t003:** Mapping between cardinal loading direction and angle locations on rotary stages. $\theta_{1}$ and $\theta_{2}$ refer to the first and second stage angles, respectively.

Loading Direction			Stage Angle (°)	
X	Y	Z	$\theta_{1}$	$\theta_{2}$
+	+	N/A	135	344
+	−	N/A	45	344
−	−	N/A	315	344
−	+	N/A	225	344
N/A	N/A	+	270	254
N/A	N/A	−	270	74

+: positive loading direction, −: negative loading direction, N/A: no loading direction.

**Table 4 sensors-22-07441-t004:** Loading cycle trials for a single loading direction. The mass of the tray was 230 grams.

Trial	Cycle	Weight + Tray (g)
1	1	0
2	1	530
3	1	1030
4	1	530
5	2	0
6	2	530
7	2	1030
8	2	530
9	3	0
10	3	530
11	3	1030
12	3	530
13	3	0

**Table 5 sensors-22-07441-t005:** Example loading cycle routines for a validation protocol. Each cycle has two rotary stage angles and a loading weight. Stated weights do not include 230 g loading tray.

$\theta_{1}$ (°)	$\theta_{2}$ (°)	W (g)
259	194	400
283	48	700
83	195	100
71	309	800
337	71	100
256	56	300
25	22	600
259	238	500
52	7	400
300	105	200
323	351	0
90	275	200
37	88	600
297	246	300
86	50	100
68	227	700
54	309	800
19	324	200
239	125	600
109	175	0
293	245	500
177	253	500
104	166	0
17	131	800
58	101	300
237	27	700
111	160	200
333	60	400
15	144	400
120	331	500

**Table 6 sensors-22-07441-t006:** Experimentally-obtained specifications for a representative 3-DOF fingertip sensor, with comparison to original requirements.

Specification	Units	Experimental Value	Requirement
Sensor mass	g	2.6	5
Range/capacity	N	14.71	20
Full scale output	V	0.289	N/A
SNR	dB	36.8	N/A
Noise	bits	6.22	N/A
Final resolution *	mN	2.564	1
	gf	0.262	0.101
Repeatability **	% FSO	0.61	5
Hysteresis **	% FSO	2.29	5
Nonlinearity **	% FSO	2.94	5
Accuracy **	% FSO	2.18	5
Root-square residual	% range	6.21	5
Dimensions	mm	23.00 × 16.18 × 10.24	N/A
Estimated unit cost	$	350	1000

* Assumes noise fluctuations are negligible after averaging filter. ** First cycle removed to avoid initial plastic zero drift effects. N/A.

**Table 7 sensors-22-07441-t007:** 3-by-3 sensitivity matrix of test sensor. For any given value, its column represents the axial component of the voltage reading being mapped from, and its row represents the axial component of the estimated weight being mapped to. For example, −180.02 in row 1, column 2 is the scaling factor applied to the Y-component of the voltage reading to get a least-squares regression estimate of the X-component of the load.

	X Voltage	Y Voltage	Z Voltage
**X load**	−2903.2	−367.68	−77.003
**Y load**	201.95	2918.9	78.264
**Z load**	−61.526	−3.7258	3943.6

**Table 8 sensors-22-07441-t008:** Coefficients and R-squared metrics of mixed-quadratic surface fits for each relative residual type.

Coefficient Type	Total Relative	X Relative	Y Relative	Z Relative
Intercept	1.93 $\times 10^{1}$	$-6.80\times 10^{2}$	$5.32\times 10^{15}$	$-1.39\times 10^{1}$
$q_{1}$	$-6.52\times 10^{-2}$	$1.26\times 10^{1}$	$-7.45\times 10^{13}$	$5.59\times 10^{-1}$
$q_{2}$	$-1.81\times 10^{-1}$	1.73	$-3.31\times 10^{13}$	$-9.93\times 10^{-2}$
$q_{1}^{2}$	$-6.35\times 10^{-5}$	$-3.85\times 10^{-2}$	$1.96\times 10^{11}$	$-1.75\times 10^{-3}$
$q_{1}\times q_{2}$	$4.69\times 10^{-4}$	$7.12\times 10^{-3}$	$1.21\times 10^{11}$	$7.09\times 10^{-4}$
$q_{2}^{2}$	$2.05\times 10^{-4}$	$-4.00\times 10^{-3}$	$-2.60\times 10^{10}$	$1.66\times 10^{-5}$
R${}^{2}$	32.16%	13.51%	7.14%	9.93%

**Table 9 sensors-22-07441-t009:** Comparative performance of 3-DOF strain sensor to existing sensors used in upper-limb rehabilitation.

Metric	Unit	Test Sensor	[4]	[5]	[27]	[28]
Sensor		Custom	FSG15N1A	MilliNewton	Nano17	Custom
DOF		3	1	1	6	5
Mass	g	2.6	3	N/A	9.07	5.93
Span	N	14.71	15	0.4 to 2	12	60
Accuracy	% FSO *	2.18	2.25	1	N/A	N/A
Hysteresis	% FSO *	2.29	3	N/A	N/A	N/A
Nonlinearity	% FSO *	2.94	1.5	N/A	N/A	1
Repeatability	% FSO *	0.61	2	1	N/A	N/A
Cost	$	350	118	N/A	5000	N/A
Planar dim	mm	23 × 16.2	12.7 × 18.2	25.4	17	9
Depth	mm	10.24	9	12.7	14.5	1

N/A: Specifications are not publicly known. * FSO: Full scale output.

## Data Availability

Publicly available datasets were analyzed in this study. This data can be found here: https://github.com/Jake-Carducci/fingersensorpaper (27 September 2022).

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
