# Peer review of "Novel Planar Strain Sensor Design for Capturing 3-Dimensional Fingertip Forces from Patients Affected by Hand Paralysis"

_sensors, 2022, doi:10.3390/s22197441_

Round 1
Reviewer 1 Report
This paper presents the design, characterization, and experimental evaluation of a novel sensor for fingertip 3D forces. The proposal is based on a planar strain gauge architecture cleverly integrated into a finger-cup interface. All aspects of the sensor proposal are detailed: design, sensing principle, FEA, data acquisition, etc. A comprehensive characterization of a first prototype is also reported showing satisfactory results on precision, accuracy, and fine sensing which suggest initial feasibility specifically for the hand rehabilitation domain.
Overall, the paper addresses an interesting topic on sensors for finger forces. The paper is well written, it is easy to read, and to follow. Yet, I recommend a major revision with the following remarks:
1. The state-of-the-art review is insufficient only with a few references.
2. Table 1 should be relocated to the Discussion section and further include the results obtained from the prototype. This way, the comparative performance is complete.
3. Fig 21 is indeed interesting but not easy to understand. There are red (or almost red) circles (mostly in the X and Z) axis suggesting 100% error load components. Is this correct? This suggests angles where applied loads should not be? A better explanation is needed.
Reviewer 2 Report
In this work, Jacob Carducci et. al. developed a novel planar strain sensor for capturing 3-Dimensional fingertip forces. As described in the article, high quality sensing of 3D forces at the fingertip requires fine resolution, low hysteresis, high precision, high linearity, resistance to fatigue effects, low package mass, and low cost. However, the technical indexes from the experimental results of the sensor developed in this paper, do not meet the expected design indexes. Compared to commercial sensors (eg, ATI Nano17), some metrics are close, some are superior, and some are deficient. This work does not well demonstrate the technical advantages and application potential of the novel sensor.
Reviewer 3 Report
The paper is very well presented and the topic is fairly interesting to readers. I only have minor comments for the authors:
1. this paper doesn't require a separate experimental section but some key information (such as the model and manufacture of the strain gauge) needs to be included in the manuscript.
2. because the gauges are not isolated from each other, there are cross-sensitivity issues. Have the authors considered modelling all 3 axis at the same time? linear regression might not be the best option here... maybe try some semi-supervised method for future works?
Round 2
Reviewer 1 Report
The paper has undoubtedly improved from its previous version. I appreciate that my remarks were taken into account, especially the more detailed explanation of Fig. 21.
I have no further major comments or remarks. Therefore, I recommend now this paper’s acceptance.
Reviewer 2 Report
In this work, Jacob Carducci et. al. developed a novel planar strain sensor for capturing 3-Dimensional fingertip forces. Agreed to accept and publish.